# Epidemiology of Mosquito-Borne Viruses in Egypt: A Systematic Review

**DOI:** 10.3390/v14071577

**Published:** 2022-07-20

**Authors:** Yuan Fang, Emad I. M. Khater, Jing-Bo Xue, Enas H. S. Ghallab, Yuan-Yuan Li, Tian-Ge Jiang, Shi-Zhu Li

**Affiliations:** 1NHC Key Laboratory of Parasite and Vector Biology, National Institute of Parasitic Diseases, Chinese Center for Disease Control and Prevention (Chinese Center for Tropical Diseases Research), WHO Collaborating Center for Tropical Diseases, National Center for International Research on Tropical Diseases, Shanghai 200025, China; fangyuan@nipd.chinacdc.cn (Y.F.); xuejb@nipd.chinacdc.cn (J.-B.X.); liyy@nipd.chinacdc.cn (Y.-Y.L.); 2School of Global Health, Chinese Center for Tropical Diseases Research, Shanghai Jiao Tong University School of Medicine, Shanghai 200025, China; j_tiane@163.com; 3Department of Entomology, Faculty of Science, Ain Shams University, Abbasiah, Cairo 11566, Egypt; eikhater@yahoo.com (E.I.M.K.); e.hamdy@yahoo.com (E.H.S.G.)

**Keywords:** dengue virus, Rift Valley fever virus, Chikungunya virus, West Nile virus, Sindbis virus

## Abstract

There are at least five common mosquito-borne viruses (MBVs) recorded in Egypt, including dengue virus (DENV), Rift Valley fever virus (RVFV), West Nile virus (WNV), Chikungunya virus, and Sindbis virus. Unexpected outbreaks caused by MBVs reflect the deficiencies of the MBV surveillance system in Egypt. This systematic review characterized the epidemiology of MBV prevalence in Egypt. Human, animal, and vector prevalence studies on MBVs in Egypt were retrieved from Web of Science, PubMed, and Bing Scholar, and 33 eligible studies were included for further analyses. The monophyletic characterization of the RVFV and WNV strains found in Egypt, which spans about half a century, suggests that both RVFV and WNV are widely transmitted in this nation. Moreover, the seropositive rates of DENV and WNV in hosts were on the rise in recent years, and spillover events of DENV and WNV to other countries from Egypt have been recorded. The common drawback for surveillance of MBVs in Egypt is the lack of seroprevalence studies on MBVs, especially in this century. It is necessary to evaluate endemic transmission risk, establish an early warning system for MBVs, and develop a sound joint system for medical care and public health for managing MBVs in Egypt.

## 1. Introduction

The spread of the coronavirus disease 2019 (COVID-19) pandemic in the past three years [1,2] has posed tremendous challenges for public health [3], hampering the control and surveillance of other infectious diseases, such as mosquito-borne diseases (MBDs) [4,5].

Egypt is located in northeastern Africa. As the connector of Asia, Europe, and Africa, Egypt belongs to the World Health Organization’s Eastern Mediterranean Region (EMR) and to the Middle East and North Africa Region (MENA). The special geographic location of Egypt aggravates the complexity and difficulty of mosquito-borne virus (MBV) surveillance and control [6]. At least five MBVs have been recorded in Egypt: dengue virus (DENV), Rift Valley fever virus (RVFV), Sindbis virus (SINV), West Nile virus (WNV), and Chikungunya virus (CHIKV). Four Rift valley fever (RVF) outbreaks occurred in the mid-to-late twentieth century and early in this century, i.e., in 1977–1978, 1993–1994, 1996–1997, and 2003 [7]. These outbreaks resulted in dramatic economic losses in the livestock industry, as well as loss of human lives. In recent decades, the expansion of live attenuated vaccines to animals has effectively slowed the spread of RVFV and curbed outbreaks [8]. In the past decade, the seroprevalence of RVFV antibodies was more than 20% in some unvaccinated livestock in both the Upper Egypt and Nile Valley governorates [9,10,11], indicating the circulation of RVFV in Egypt. Moreover, RVF outbreaks occurred continuously in neighboring countries, such as Sudan in 2007, 2010, 2011–2012, and 2019 [12] and Kenya in 2006–2007 [13].

Dengue outbreaks in Egypt were reported in 1799 in Cairo and Alexandria [14]. In 1937, 2594 human cases were recorded in Cairo [15]. DENV has been largely controlled in Egypt by focusing on the eradication of its mosquito vector, *Aedes aegypti*, following the introduction and intense usage of the DDT insecticide [16]. In 2011, DENV was unexpectedly detected in two Italian tourists returning from South Egypt [17]. In 2015, a DENV resurgence occurred in the Dairoute District of the Assyoute (Assiut) Governorate, with at least 253 cases [18,19]. In 2017, two cases of DENV were reported in travelers returning to Moscow, Russia, from Hurghada, Egypt, on the Red Sea coast [20]. In the same year, a DENV outbreak with at least 680 cases occurred in the Red Sea Governorate, where the reintroduction and breeding of *Ae. aegypti* were confirmed [21]. In the last decade, DENV outbreaks have occurred in the Red Sea region, Yemen, Sudan, Djibouti, and Saudi Arabia [22,23].

In addition to RVFV and DENV, sporadic cases or seropositivity for WNV [24,25], SINV [26], and CHIKV [27] have been reported in Egypt. The occurrence of unexpected disease outbreaks and occasional exported cases indicate that undetected DENV/WNV transmission occurred in Egypt before these events. These occurrences also reflect the deficiencies of the MBV surveillance system in Egypt, especially the absence of a system for early warning. In this article, we present a systematic review of the historical records of MBVs in Egypt to characterize the epidemiology of MBVs, with the aim of achieving evidence-based and informed risk prevention and control of these viruses and the diseases they cause.

## 2. Methods

### 2.1. Data Sources and Search Strategy

A systematic literature search for relevant articles was performed according to the PRISMA criteria [28]. We performed an electronic literature search in the databases of Web of Science, PubMed, and Bing Scholar, using different combinations of the following keywords: “Egypt” and “Mosquito-borne virus, mosquito-borne diseases, MBV”, or “West Nile virus, West Nile fever, WNV”, or “dengue virus, dengue fever, DENV”, or “Rift Valley fever virus, Rift Valley fever, RVFV”, or “Sindbis virus, Sindbis fever, SINV”, or “Chikungunya virus, Chikungunya fever, CHIKV”. Articles published from the database inception to 28 May 2022 were included in this study, if they fulfilled the following selection criteria: (i) that they were written in English or had an English-language abstract; (ii) that they pertained to the isolation or detection of MBVs from mosquito vectors or hosts by a reverse transcription-polymerase chain reaction (RT-PCR); or (iii) that they pertained to testing for the presence of MBV antibodies in the host by serological analyses. Additional articles were selected by screening the references of papers that met our inclusion criteria. The following exclusion criteria were applied to titles, abstracts, and full texts: (i) that they related to mosquito-borne parasites; (ii) that they related to target disease control, surveillance, and evaluation/assessment of laboratorial detection capability; (iii) that they consisted of a case report, a clinical study, a study in which cases were observed in returned travelers, or a systematic review; (iv) that they related to virus ultrastructural observations, laboratory susceptible experiments, and phylogenetic studies; (v) that they related to a vaccine study; (vi) that they related to a comparison of methods for MBV detection; or (vii) that the study area was located outside Egypt.

### 2.2. Data Extraction

Data were extracted from the selected studies using a researcher-made and -piloted data extraction form in Excel. Eligible studies were compiled by virus, organized by year, and then stratified by subjects’ categories as follows: (1) for studies on human and animal subjects, we extracted data based on the year of implementation, the city/governorate, the sample size, the age and sex of participants (for human subjects only), the species of animal, the laboratory methods, and the estimated assay-based MBV prevalence; and (2) for studies on vector populations, further data were extracted, including information on vector species, the sample size of the species (vectors) tested, and the number of positive pools for each species.

### 2.3. Risk of Bias Assessment

To assess the quality of the eligible studies identified through the systematic review, the risk of bias (ROB) was assessed for each study using the Cochrane approach [29] and by evaluating the precision of the reported measures. Each MBV prevalence measure was considered to have a low or high ROB in two domains: sampling methodology and type of assay used for MBV detection. ROB was considered low for the following conditions: (i) sampling that was based on probability (i.e., using random or cluster selection) and (ii) MBV prevalence measures that included viral neutralization testing for general seroprevalence studies or biological assays (i.e., virus isolation, RT-PCR) for acute infection studies. ROB was considered high for (i) acute infection studies or studies on sera or tissues taken from affected herds and (ii) MBV prevalence measures that included an agar gel precipitation test, a complement fixation test, an enzyme-linked immunosorbent assay (ELISA), a hemagglutination inhibition test, and an immunofluorescence assay. Studies were considered to have high precision if the number of humans or animals tested was greater than 100 or if more than 1000 for mosquitoes were involved [30,31].

## 3. Results

### 3.1. Search Results

The selection process, based on PRISMA guidelines, is shown in Figure 1 [32]. The search yielded 367 reports, 325 of which were initially screened after the removal of duplicates. Following the screening process, 33 of the 325 studies, covering RVFV, DENV, WNV, SINV, and CHIKV, were eligible for inclusion in the study (Table 1, Figure 1). Sixteen of the 33 studies involved human samples. Nineteen, two, and three studies were performed on domestic animals (camels, sheep, cattle, buffalo, goats, pigs, horses, and donkeys), wild animals (mice, rats, and birds), and vectors (mosquitoes and ticks), respectively. General information on geographic distribution, host, vector, and known genotype of MBVs that have been historically recorded in Egypt is provided in Table 1.

### 3.2. Risk of Bias Assessment Results

A summary of the ROB assessment results is presented in Table 2. In brief, most studies (25 out of 33) had the sample size required to yield high precision. Among the included human studies, four were performed on the blood samples of patients with a history of fever. Other studies utilized random or cluster sampling, including one study that was performed on blood donors; hence, they had a low ROB. Tests such as virus isolation, neutralization tests, or RT-PCR were performed in 10 prevalence studies, entailing a low ROB for the assays used.

### 3.3. MBV Prevalence among Humans, Animals, and Vectors in Egypt

#### 3.3.1. DENV

Three human prevalence studies for DENV were identified. Seroprevalence in healthy male university students was 0.3% in 1969 [33] and was absent in the serum samples of children with acute febrile illness in 1968 in Alexandria [26]. This gap in the literature has existed for half a century. In Sohag and Assiut in 2019, the positive rates of DENV antibodies in the general population aged <21 years, 21 to 40 years, and >40 years old were 25%, 11.32%, and 10%, respectively, and 3.30% in local camels [34].

#### 3.3.2. RVFV

Nineteen studies on RVFV prevalence were eligible for inclusion in the systematic review, 58% of which were conducted before the year 2000. Figure 2 illustrates the geographic distribution of prevalence studies on RVFV in Egypt. Of the prevalence studies, 84.2% were conducted on domestic animals and 57.9% involved human samples. Over 30% of the camels sampled at the southern borders of Egypt were seropositive for RVFV antibodies, according to an investigation of the 1977 RVF outbreak [61]. RVFV prevalence was comparatively even in the decade after the 2003 RVF epidemic. In the Nile Valley governorates, the seropositivity rate of RVFV was <10% in domestic animals born after the RVF epidemic in 2003 [10,11]. In a random community-based study published in 1993 on the human prevalence of RVFV in Nile Valley governorates, the positivity rate of the RVFV antibody was 14.97% [41]. The most recent human prevalence study on RVFV was performed in suspected cases of RVFV in 2003, with positivity rates of 4.00% and 5.06% by virus isolation and RT-PCR, respectively [50]. In slaughterhouse personnel, the RVFV antibody positivity rate was 21.2% in 1999 [46] and 13.95% in 2010 [51], as tested by ELISA. More recently, between 2013 and 2015, the seropositivity rate for RVFV antibodies in unvaccinated cattle in the Nile Valley governorates was 26.9% (36/134) [11]. In domestic animal serum samples collected from 2018 to 2019 from the southern parts of Egypt, the RVFV antibody positivity rate in sheep was as high as 65.21%, compared with the RVFV antibody positivity rate in camels of 20.65% [9].

RVFV was isolated from rats (one out of eight) [35], although RVFV antibodies were not observed in wild rodents during the 1977–1978 RVF epidemic in Sharqiya, Egypt [61]. A positivity rate of RVFV antibodies of approximately 30% was detected in 300 tested blood samples of the wild black rat *Rattus rattus* in the Behera governorate around the year 2000 [48]. *Cx. pipiens* [62] and *Cx. antennatus* [50] are the primary vectors of RVFV in Egypt, as confirmed by virus isolation tests based on wild samples, while *Ae. caspius* was suspected to transmit RVFV in livestock animals [63].

#### 3.3.3. WNV

A total of 11 seroprevalence studies of WNV were identified from the eligible reports. WNV, first detected in Egypt, can be traced back to 1951. It was isolated from three serum samples of 251 children with a history of fever who lived in a community that is 30 km away from Cairo [64]. The prevalence of the WNV antibody was 3% (15/437) among school children aged 8 to 14 years living in the Nile Valley governorates in 1989 [40]. During the period from 1999 to 2002, the human seroprevalence rates for the WNV antibody ranged from 1% to 35% in Upper Egypt, Middle Egypt, Lower Egypt, North Sinai, and South Sinai, while the seroconversion rates were 18%, 17%, and 7% in Upper Egypt, Middle Egypt, and Lower Egypt, respectively [57]. Notably, 49% of seroconverters reported an undiagnosed febrile illness during the studied years [57]. In 1999, the WNV antibody had the highest seropositivity rate (143/264, 54.14%) among MBVs in Egyptian workers at sewage treatment plants (STPs) [47]. Recently, the sera collected between 2013 and 2014, mainly from the rural areas of the Nile Valley governorates, of more than half of blood donors (88/160, 55%) were seropositive for WNV IgG antibodies [58].

The WNV antibody positivity rates in adult horses ranged from 14% in Alexandria (northern Egypt) to 89% in Qena (Upper Egypt), according to a neutralization test carried out in 1959 [54]. In a study conducted in 2018, 16.8% of 930 horses from five governorates in the Nile Valley were serologically positive for WNV, with the highest seroprevalence in horses aged more than 15 years [65]. In the same year, the seroprevalence rates for WNV in other domestic livestock in the Nile Valley governorates (Qalyoubiya, Menoufiya, Kafr El-Sheikh, and Gharbiya) were 18% (18/100), 0% (0/50), 40% (20/50), 3.5% (3/85), and 5.3% (4/75) in cattle, buffalo, camels, sheep, and goats, respectively. [65]. For chickens, the seroconversion rate for the WNV antibody, detected using ELISA, ranged from 4% in Lower Egypt to 47% in Middle Egypt during the period from 1999 to 2002 [57]. One out of 25 migrating storks captured over the Sinai Peninsula during the autumn of 1998 and the spring of 1999 was positive for the WNV antibody [66].

*Culex pipiens*, *Cx. antennatus*, and *Cx. perexiguus* were identified as vectors of WNV in Egypt [56,60]. In addition to mosquitoes, the natural infection of WNV in ticks (*Argas reflexus hermannii*) was found in pigeon cotes located in Kafr El-Sheikh in 1960 [55].

#### 3.3.4. SINV

Four prevalence studies for SINV in Egypt were eligible for inclusion in the systematic review, and all were conducted before 2000. In 1968, SINV antigens were detected by hemagglutination inhibition and complement fixation in 4% acute sera from children living in Alexandria [26]. In 1969, the seropositivity rate for SINV was 6% among 1113 male university students [33]. In 1999, the rate was 1.13% (3/264) among Egyptian workers in STPs [47].

In 1952, SINV was first isolated from *Cx. pipiens* and *Cx. univittatus* samples collected from a village located a few kilometers away from Cairo [67]. In mosquito samples collected from Aswan in 1993, three pools out of 9011 individuals of *Cx. perexiguus* were determined to be positive for SINV by virus isolation tests, whereas no SINV was found in *Anopheles multicolor*, *An. pharoensis*, *An. tenebrosus*, *Cx. antennatus*, *Cx. pipiens*, *Cx. poicilipes*, *Ae.* (=*Ochlerotatus*) *caspius*, or *Urranotaenia unguiculata* [56].

#### 3.3.5. CHIKV

In 1984, the seropositivity rate for the CHIKV antibody was 5.45% in 55 blood samples of humans with acute fever in Giza [24]. No studies on the potential mosquito vectors for CHIKV have yet been reported in Egypt.

## 4. Discussion

### 4.1. DENV Resurgence and Potential on (Implications for) Continuous Outbreaks

Records of DENV outbreaks in Egypt can be dated back to 1799, when such outbreaks occurred in Cairo and Alexandria [14]. From September to November of 1937, at least 2594 human cases and 50 deaths occurred in Cairo [15]. The disease burden of DENV in Egypt has declined sharply, due to the eradication of *Ae. aegypti* after the introduction of DDT to Egypt in the 1940s [16]. Among human prevalence studies for DENV, the seroprevalence rate for DENV antibodies among university students from widely differing localities was 0.3% in 1969 [33]. DENV reemerged in Egypt, as evidenced by febrile illnesses in two Italian tourists who traveled back from southern Egypt in 2009 and 2010 [17]. In 2015, an outbreak of dengue fever occurred in Daiyroute District, Assiut governorate (Upper Egypt) [18] that was similar to the 1937 DENV epidemic in Egypt. In 2017, up to 2500 human cases were recorded in the Red Sea Governorate, where *Ae. aegypti* was reintroduced and reestablished [21]. Two cases of dengue fever were reported in travelers returning to Moscow, Russia, from the Red Sea coast of Egypt in the same year [20]. In 2019, the seropositivity rate for DENV antibodies in the population under the age of 21 years was 25%, although the rate was determined based on a limited sample size (*n* = 8) [67]. Multiple studies on DENV seroprevalence in the Red Sea region and Pakistan have indicated a rate of more than 20% DENV seroprevalence in the general population [23]. Moreover, DENV has been prevalent in camels in Egypt, with an antibody positivity rate of 3.3% [34], implying that camels may serve as reservoirs of DENV. The seropositivity rate for DENV was 47.6% among residents from El-Gadarif State, Sudan [68]. The largest number of dengue cases ever reported globally was in 2019; in Sudan and Yemen, the number of dengue fever cases was even higher in 2020 (https://www.who.int/news-room/fact-sheets/detail/dengue-and-severe-dengue, accessed on 28 May 2022). Thus, the potential disease burden of DENV both in Egypt and in neighboring countries indicates that more attention should be paid to DENV case surveillance and vector control in Egypt. The serotype of DENV identified in the 2015 DENV outbreak in Egypt was serotype I, according to ELISA and PCR [19]. To the best of our knowledge, there is no available DENV genomic sequence in GenBank or in the literature. In other words, it is unclear which dengue genotypes circulate in Egypt, in both past and recent outbreaks; therefore, DENV genome sequencing is essential to trace the source of dengue strains involved in DENV outbreaks in Egypt.

### 4.2. Seroprevalence Recovery and Frequent Livestock Importation Are the Main Factors for RVFV Resurgence

Rift Valley fever is an arthropod-borne viral zoonotic disease that affects wildlife, livestock, and humans [69]. In cattle, RVF seroprevalence was extremely high (55.56%, 5/9) during the 1993 RVF epidemic in Aswan (Upper Egypt) [44]. In contrast, it was less than 10% prior to and after 1993. RVFV studies on horses were rare, with a seroprevalence rate of 5.56% in Aswan in 1993 [44]. In sheep, it was 41.80% in the Nile Valley (Qalyoubia and Menia) during the 1977 RVF epidemic [35]. In the period between 1980 and 2000, RVFV in sheep was circulating in the Nile Valley governorates at a low prevalence rate (1.20% to 5.91% during the period from 1984 to 1986) [38,39]. In contrast, the seroprevalence rate for RVFV antibodies in sheep was 34.48% in Aswan in 1993 [44]. This rate decreased to below 5% in the Nile Valley (Kafr El-Sheikh, Giza) in the first decade of this century [50,52]. However, recently, the sheep seroprevalence for RVFV antibodies was significantly different in the Upper Egypt and Nile Valley governorates, at 39.67% and 0.72%, respectively [9,10], which was probably due to continuous livestock importation from Sudan into southern Egypt [70,71]. Two studies on RVFV prevalence in camels before 2010 showed that RVFV was low in camels from Giza, even during the 1977 RVF epidemic [35,52]. However, in recent decades, camel seropositivity rates for RVFV antibodies in both the Nile Valley and Upper Egypt governorates were greater than 20% [9,53]. In addition, in a survey published in 2009, the seropositivity rate was 15.10% in pigs from Alexandria (northern Egypt) and 13.95% in humans who are in frequent contact with pigs [51]. In comparison, the rate was under 5% in the general human population in the twentieth century [24,40,69]. RVFV isolated from *R. rattus frugivorus* was collected in Sharqiya during the 1977 RVF epidemic, with no clinical manifestations [35], although RVFV antibodies were absent in wild rodents in the same governorate in 1977 [61], suggesting that mice could be a natural reservoir of RVFV [35]. Furthermore, a study published in 2001 showed that the seropositivity rate for RVFV antibodies in *R. rattus* was 29.33% in the Nile Valley governorates [48,49].

RVFV strains detected in mosquitoes, humans, and domestic animals in Egypt during the 1977–1978 epidemic were monophyletic, belonging to lineage A, with very limited genetic diversity, indicating a single introduction that was probably from central Africa in around the 1950s [72,73]. Moreover, the 95EG Cow-2509 strain, from the aborted fetus of a cow after vaccination with the 95EG vaccine in Egypt, which was grouped in lineage L, was distant from previous isolates from Egypt in lineage A but closer to strains that were prevalent in Kenya, Zimbabwe, and South Africa in the 1970s, based on the M segment [74,75]. Phylogenetic studies of RVFV strain prevalence in this century are rare. There were a few unpublished sequences obtained in ticks (HQ403562–HQ403564, M segment) and camels (MN395695–MN395697, S segment) in GenBank. Thus, we generated a phylogenetic tree (Appendix A), based on the S segment, to determine whether the circulating RVF genotypes have changed in Egypt in recent decades, and to determine whether the tendency toward increases in the RVFV antibody positivity rate was induced by the introduction of RVFV strains from neighboring countries or southern African countries. In the tree, camel sera from both the Red Sea in 2013 (MN395697) and Aswan (Upper Egypt) in 1994 (EU312110) clustered within lineage A, which included strains that were obtained in RVFV epidemics in Egypt during 1977 and 1978 and the SH272655 strain (KY366331) that came from Mauritania in 2015. This indicated that RVFV lineage A permanently circulates in Egypt. It is worth noting that the SH272655 strain was detected from blood samples of suspected cases involved in the 2015 RVF outbreak in Mauritania, with 57 confirmed cases and 12 deaths. This was the first evidence that the strains were prevalent in Northeastern African and circulating in Mauritania [76].

Although *Cx. pipiens* and *Cx. antennatus* are the primary vectors of RVFV [8], *Ae. caspius*, among the mosquitoes collected in the 1993 RVF epidemic area, appeared to be the most efficient vector of Egyptian mosquitoes that were evaluated under laboratory conditions [77]. Additionally, *Ae. caspius* prefers to feed on humans, bovines, and equines [78]. Thus, *Ae. caspius* is a suspected vector of RVFV in Egypt. Five common African mosquito species, *Ae. palpalis*, *Ae. circumluteolus*, *Ae. mcintoshi*, *Cx. antennatus*, and *Cx. pipiens*, were laboratory-confirmed as transmitting RVFV by bite after oral exposure [79]. In the RVF outbreak in Sudan in 2007, RVFV was detected in all three common mosquito genera, *Culex*, *Anopheles*, and *Aedes*, and low rates of RVFV positivity were found in the larval and/or male stages of *An. arabiensis*, *An. coustani*, *Cx. pipiens* complex, and *Ae. aegypti*, indicating transovarial and venereal transmission of the virus within these mosquito species [80].

There are three available veterinary-licensed anti-RVF vaccines, including the Smithburn live attenuated virus vaccine, an inactivated virus vaccine, and the clone-13 live attenuated virus vaccine [81]. An animal vaccination scheme was applied in Egypt during the 1993–1994, 1996–1997, and 2003 outbreaks, especially in imported livestock [82]; it was also used in domestic livestock farms [11]. The sharp decrease in human cases in the last three RVF epidemics in Egypt was probably due to the effectiveness of vaccination [7]. However, adverse effects, such as abortions and liver and kidney necrosis, occurred in some livestock [74,83]. Moreover, vaccination has some limitations, as the vaccines cannot be injected into sheep and goats under 4 months of age, and vaccinated animals cannot be slaughtered for human consumption within 21 days post-vaccination [82]. In fact, the coverage of RVF vaccines is unclear, especially for livestock belonging to small holders (owners).

The main risk factors and bottlenecks in controlling potential RVF outbreaks in Egypt are the sustainable circulation of highly contagious RVFV strains in Egypt, the increased seropositivity rates for RVFV antibodies among hosts, the sporadic RVF outbreaks in East Africa and the Middle East [84], continuous livestock trading with neighboring RVF endemic countries, and the lack of a licensed vaccine against RVFV for use in humans.

### 4.3. Robust WNV Circulating and Extremely High Seroprevalence in Humans

West Nile virus has been detected in more than 60 mosquito species in the Eastern Mediterranean region [80]. In Egypt, WNV was isolated from *Cx. pipiens*, *Cx. antennatus*, and *Cx. perexiguus*, which were collected in Aswan during the 1977–1978 RVF outbreaks [56]. WNV was also detected in a few *Culex* spp. in 2002 [57] and 2019 [60], and in ticks in 1960 [55]. WNV was positive in one of the 25 exhausted storks that were captured over the Sinai Peninsula during the southward migration in 1998 [66]. Egypt is on the flyway for migratory birds in the Black Sea/Mediterranean and in East Africa/West Asia, suggesting a potential role of migratory birds and indigenous wild birds in WNV transmission in Egypt [85,86].

The seropositivity rate for WNV in humans continuously increased in the second half of the last century and affected more than half of the workers at STPs [47] and blood donors [58]. The seroconversion test for WNV in humans from Upper, Middle, and Lower Egypt, conducted at the beginning of the 2000s, showed that 15% of individuals, on average, were positive for WNV antibodies in a single year, which indicated that WNV was actively circulating during this period [57]. In 2012, both Germany and Holland reported imported WNV cases from Egypt [25,87]. In general, just as it was in humans [57], the WNV antibody seropositivity rates in chickens [57] and equines [54] from Upper Egypt were higher than the rates from Nile Valley governorates.

The WNV prevalence in Egypt belongs to lineage 1-a [59,88]. The time span of the two available WNV sequences from Egypt is almost 70 years, but they have shown very limited genetic diversity; they are most closely related to each other in the phylogenetic tree [59]. This supports the speculation that frequent transmission of WNV occurred in Egypt. Although no historical WNV outbreak has been recorded in Egypt, WNV lineage 1-a includes isolates from Africa, Europe, the Middle East, Asia, and the Americas, and it has been associated with significant outbreaks in humans in the USA, Russia, Italy, and China [89,90]. Low morbidity and hospitalization, along with high seroconversion to WNV, indicate that WNV in Egypt is widely prevalent, but it is probably a relatively mild disease [57].

### 4.4. The SINV Strain, Once Prevalent in Egypt, Was Phylogenetically Close to the Strains Involved in the SINV Outbreak in Northern Europe

The SINV supposedly originated in Africa, then was spread to Eurasia and Oceania by migratory birds; however, clinical SINV infections and outbreaks have been reported mostly in northern Europe, especially in Finland [91,92]. In 1968, SINV antigens were found in 4% (5/120) of undiagnosed fever cases among children living in Alexandria [26]. In the middle of the twentieth century, human seroprevalence for SINV in 1113 healthy male participants in Egypt was low (6%) [33]. At the end of the 1990s, the seropositivity rate of SINV antibodies was 1.13% among workers who were potentially at risk of infection [47]. No data on SINV prevalence in Egypt have been published in this century. *Cx. pipiens*, *Cx. univittatus*, and *Cx. perexiguus* were vectors of SINV in Egypt, as indicated by virus isolation tests [56,67]. Like WNV and SINV can be transported by migratory birds over long geographical distances [93]. The typical symptoms of SINV are arthritis, itching rash, fatigue, mild fever, headache, and muscle pain [94]. The Egyptian SINV AR-339 strain, isolated from *Cx. pipiens* and *Cx. univittatus*, was phylogenetically close to strains that were isolated from mosquitoes in Kenya in 2013, belonging to genotype I [67,95]. The SINV genotype I cluster contains strains that were obtained from the 2002 Finland SINV outbreak [94], as well as other European SINV strains [95]. It was speculated that the AR-339 p270 strain isolated from Egyptian mosquitoes, which was collected in 1952, was close to the ancestor of SINV genotype I [91], and that is widely distributed, spreading to Central and South Africa, northern Europe, and Southeast Asia [94,95,96].

### 4.5. Few of Seroprevalence Studies on CHIKV in Egypt and Severe CHIKV Outbreaks in the Red Sea Region

Chikungunya virus was thought to have originated from central Africa within the last 500 years, with Asian strains circulating primarily in the urban *Ae. aegypti*/*Ae. albopictus*-human transmission cycle, and the African CHIKV circulating primarily in the *Ae. furcifer*/*Ae. africanus*-primates sylvatic/enzootic cycle [97]. However, in this century, the “sylvatic/enzootic”-originated African strains (eastern/central/southern African genotype) have demonstrated spillover and a capacity to circulate in the urban cycle, being involved in outbreaks in Africa, including Congo (2000), Kenya (2004), Gabon (2006–2007), Yemen (2011), Djibouti (2011), Sudan (2005; 2018), and India (2006) [98,99,100,101,102,103,104,105,106]. Eastern Sudan experienced the largest CHIKV outbreak in Africa, affecting approximately 500,000 people, with a great incidence of severe illness, fatality, and long-term disability [99]. To the best of our knowledge, there has been only one seroprevalence study on CHIKV in Egypt, which was conducted in Giza in 1984, in which the seropositivity rate for CHIKV was 5.45% (3/55) in 55 human blood samples with non-specific fever and myalgia. As no CHIKV was recorded in Egypt before 1984, with CHIKV isolated in several African countries, patients with CHIKV antibody positivity may have acquired the infection outside Egypt [24].

Dengue virus and CHIKV share not only the (transmitting) primary mosquito vectors, but also the dominant factors that characterize both DENV and CHIKV outbreaks, such as high population density and easy cross-border movements among nations with outbreaks [107]. The CHIKV has been active in the Red Sea and Indian Ocean regions in recent decades [27,97]. In light of the presence/reemergence of *Ae. aegypti*, the resurgence of DENV epidemics, severe CHIKV outbreaks in neighboring countries, and almost no studies of CHIKV prevalence, there is a necessity to conduct a baseline survey on human seroprevalence of CHIKV antibodies in Egypt, thereby complementing, integrating, and enhancing the vector surveillance system for DENV and CHIKV.

One limitation of this review is that a few of the eligible studies involved in the analyses were only abstracts that were mainly published in the 1970s and 1980s. Thus, relevant information on the sample selections and the specific assays from these studies is unavailable for a further epidemiological survey. However, our study reflects the deficiency of MBV studies and underscores the necessity for improved systematic MBV surveillance in Egypt.

## 5. Conclusions

Mosquito-borne viruses are undoubtedly a growing public concern in Egypt. In view of the seroprevalence studies on DENV, RVFV, and WNV, the spread and infection rates of the three MBVs have increased in recent years. Due to the reintroduction and established breeding of *Ae. aegypti*, two DENV outbreaks have occurred in Egypt in the past decade [19,21]. At the same time, *Ae. aegypti* range expansion increased the transmission risk of CHIKV in Egypt, which may be worsened by the CHIKV pandemic in Sudan [99] and other Red Sea countries [27].

Rift Valley fever virus prevalence declined to a low frequency in natural circulation after the 2003 RVF epidemic, but the seroprevalence recovered in recent years, especially in Upper Egypt [9]. Notably, the RVFV strains retrieved in Egypt, spanning almost half a century, are monophyletic. By coincidence, the two available WNV strains with genomic information from Egypt, obtained in 1951 and 2020, are most closely related to each other in phylogeny [60]. This indicates a robust circulation and frequent transmission of both RVFV and WNV in this country. In addition, exported cases of DENV and WNV from Egypt to other countries have been recorded [17,20,25,94]. Moreover, the RVFV strains involved in the 2015 Mauritania RVF outbreak clustered in the lineage that mostly contained those from Egypt [76]. The common drawbacks of MBVs surveillance in Egypt are the lack of seroprevalence studies on MBVs, particularly for CHIKV, and the lack of genome data on MBVs from positive serum and vector samples, especially in this century. These drawbacks inhibit the acquisition of evidence on transmission routes and the extent of the burden of the disease. Due to the lack of current available evidence to characterize the prevalence of MBV at home and the active and occasional (sporadic) outbreaks that occur in neighboring countries, Egypt is probably on the border of the MBV storm, or even near the center. Thus, it is essential to estimate an accurate disease burden, to evaluate the risk of endemic transmission of MBVs, and to establish an early warning system by conducting the following exercises: (1) host seroprevalence studies, especially in high-risk populations; (2) systematic surveillance of mosquito population diversity, distribution, seasonal abundance, pathogen infection rate, source tracing, and vector species distribution, as well as insecticide resistance levels for early warning and vector control; and (3) regular and random pathogen detection in imported/domestic livestock, wildlife, and migratory birds of the presence of MBVs and other zoonotic diseases. There is also an urgent need to establish a sound joint system for medical care and public health in relation to MBVs via the following steps: (1) conducting health education on personal protection against mosquito bites and increasing awareness of MBVs prevalence in Egypt; (2) popularizing and strengthening the diagnosis and treatment skills of clinicians with respect to MBVs; and (3) establishing a direct network system for reporting MBVs and following the dynamics of the disease burden on a timely basis. Concerted synchronized efforts among government organizations, research institutions, healthcare systems, and the public will help in planning effective prevention of outbreaks and in developing epidemic disposal strategies against threats from MBVs in Egypt and neighboring countries.

## Figures and Tables

**Figure 1 viruses-14-01577-f001:**
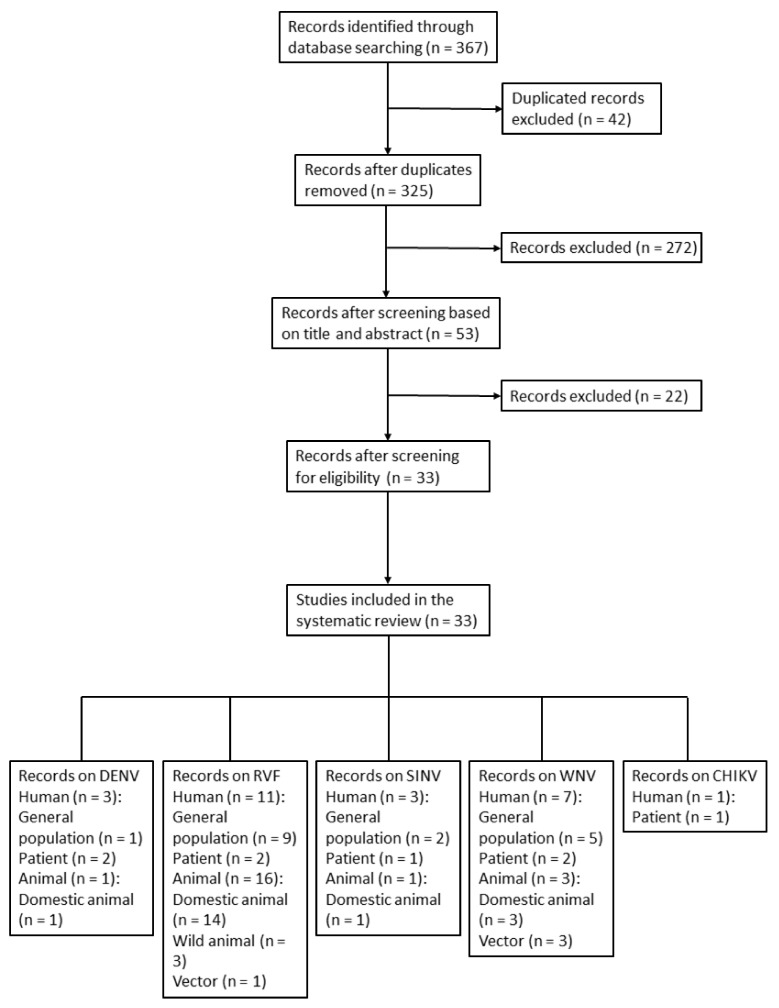
Flow diagram of article selection for prevalence of mosquito-borne virus in humans, animals, and vectors in Egypt. The study categories were subdivided accordingly. Humans: general population and patients; animals: domestic animals and wild animals.

**Figure 2 viruses-14-01577-f002:**
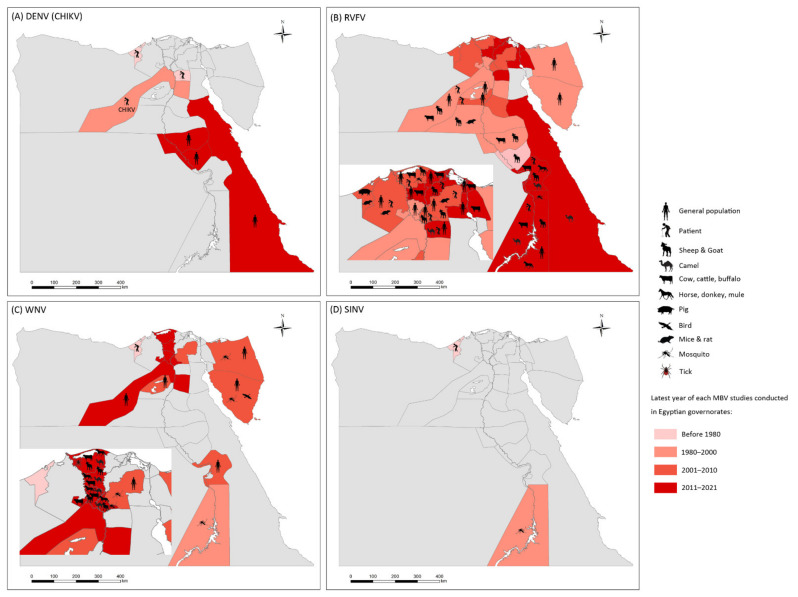
Epidemiology of mosquito-borne viruses reported among vector, animal, and human populations in Egypt. (**A**) dengue virus (DENV) and Chikungunya virus (CHIKV); (**B**) Rift Valley fever virus (RVFV); (**C**) West Nile virus (WNV); (**D**) Sindbis virus (SINV).

**Table 1 viruses-14-01577-t001:** Summary of seroprevalence studies for mosquito-borne viruses in Egypt.

Virus	Year, Ref.	Region/City/ Governorate		Species	Sample Size	Participants Characteristics		Assay	Elisa (IgG)		VI	NT	IF	HAI	CF	RT-PCR	AGPT	Remarks
						Male/ Female	Age Range		Male/ Female	Age Range								
DENV	1968, [26]	Alexandria		Human	120	60 60	3–13	CF HAI						0.00	0.00			Acute blood samples from children
	1969, [33]	Throughout Egypt		Human	1113	1113 (male)		HAI						0.30				University students Abstract
	2019, [34]	Upper Egypt (Sohag and Assiut)		Human	91	57 34	8 (<21) 53 (21–40) 30 (>40)		14.03 8.82	25.00 11.32 10.00								70 individuals with breakbone fever
				Camel	91	76 15			2.63 6.67									
RVFV	1977, [35]	Qalyoubiya Sharqiya Giza Menia Aswan Sohag Cairo		Sheep Cow Camel Goat Horse Rat	58 17 30 9 18 8			VI			46.55 5.88 3.33 11.11 5.56 12.50							Sera or tissue from sick feverish and died herds
	1977, [36]	Qalyoubiya Sharqiya Giza		Human	91			VI			59.34							Acute febrile and/or fatal haemorrhagic
		Sharqiya		Sheep	4			VI			50.00							Aborting sheep
	1977–1978, [37]	Sinai Peninsula		Human	170			HAI						4.71				Abstract Swedish soldiers serving in Egypt
	1984, [38]	Nile Delta		Sheep	406			HAI NT				5.91		5.91				Abstract
	1984, [24]	Giza (from Cairo or surrounding areas)		Human	55			VI HAI			0.00			0.00 (IgM) 1.82 (IgG)				Serum samples of non-specific fever and myalgia patient
	1986, [39]	Daqahliya		Sheep	1714			ELISA (IgM) HAI NT	0.00			1.17		1.17				Unvaccinated Abstract
	1989, [40]	Sharqiya		Human	223	99 124	8–14	ELISA	3.03 4.03									School children
	NA (1993 ^a^), [41]	Nile River Valley		Human	915			ELISA	14.97									Abstract
	1993, [42]	Behera Beni Suef Cairo Damietta Daqahliya Gharbiya Giza North Sinai Ismailia Kafr El-Sheikh Menoufiya Port Said Qalyoubiya Sharqiya South Sinai		Human	46 65 100 100 99 100 68 47 100 100 100 99 100 26 31			ELISA (IgM)		0.00 0.00 0.00 1.00 2.02 3.00 0.00 0.00 10.00 1.00 2.00 3.03 0.00 7.69 0.00								Abattoir workers
	1993, [43]	Aswan	Village 1	Human				ELISA	11.96									IgM
			Village 2						8.36									
	1993, [44]	Aswan		Sheep Goat Cow Buffalo	22 7 8 1			IF					36.36 28.57 50.00 100.00					
	1997, [45]	Aswan and Assiut		Sheep Cattle	57 93			NT, IF, CF, AGPT				100.00 100.00	100.00 100.00		100.00 100.00		84.21 74.19	Aborted sheep fetuses and sera from affected herds
	1999, [46]	Egypt		Human	52			ELISA	21.15									Slaughter house personnel in Makkah during Hajj 1419 (1999)
	1999, [47]	NA		Human	264			ELISA	7.95									Worker of sewage treatment plants Abstract
	NA (2001 ^a^), [48,49]	Behera Alexandria Menia		Rat				ELISA, IF, RT-PCT	29.33				6.00			9.67		Abstract
	2003, [50]	Aswan, Beni Suef, Behera, Cairo, Daqahliya, Kafr El-Sheikh, Qalyoubiya, Qena, and Sharqiya		Human	375			VI, RT-PCR			4.00					5.06		Sera or cerebrospinal fluid from suspected cases of RVF
		Kafr El-Sheikh		Cow Sheep Goat Buffalo	48 36 9 8			ELISA (IgM)		10.42 5.55 0.00 0.00								
		Kafr El-Sheikh		*Culex antennatus* *Cx. pipiens* *Cx. perexiguus* *Ae. detritus* *Anopheles tenebrosus* *An. pharoensis*	8798 (218) 102 (27) 6 (3) 1 (1) 248 (41) 24 (17)			RT-PCR								1.38 0.00 0.00 0.00 0.00 0.00		
	NA (2009 ^a^), [51]	Alexandria		Pig	245			ELISA, HAI	15.10					8.16				Abstract
				Human	43			ELISA HAI	13.95					6.98				Veterinarian and their assistants, butchers and abattoir workers
	2009, [52]	Giza (originated from governorates throughout Egypt)		Cattle Buffalo Sheep Camel	161 153 174 10			ELISA	1.24 3.26 0.00 0.00									
	2013–2015, [11]	Nile Delta (Damietta, Port Said, Daqahliya)		Cattle (immunized)	2743		2–7	ELISA IF NT	16.00			58.25	66.84					Farms
		Nile Delta (Daqahliya, Gharbia, Damietta, Port said)		Cattle (non-immunized)	1289		2–7	ELISA IF NT	7.29			26.87	56.76					Small holders
	2014–2015, [10]	Daqahliya Daqahliya Daqahliya Ismailiya Cairo Red Sea		Sheep (small holder) Goat Buffalo (small holder) Buffalo (farms) Camels (local abattoir) Camels imported from Sudan	438 26 88 100 71 150		2–10 2–3 3–7 4–6 5–7 2–7	ELISA NT IF	0.72 0.00 10.26 1.05 100.00 0.00			0.00 0.00 3.57 0.00 10.17 0.00	3.41 NA 25.00 0.00 2.82 0.00					Non-vaccinated (born after RVF epidemic in 2003)
	NA (2018 ^a^), [53]	From different herds in Egypt and originated from Sudan		Camel	200 (120: origin from Sudan 80: local breed)	70 130	≤1 2–5 5–9 ≥9	ELISA	21.43 9.23	19.23 3.33 13.93 25.71								
	2017–2019, [9]	Aswan, Qena and Luxor		Cattle Sheep Goat Camel Donkey	92 92 92 92 92		0.5–3	ELISA	5.55 65.21 14.44 20.65 0.00									Unvaccinated
WNV	1959, [54]	Nile Delta and Upper Egypt		Horse Donkey mule	102 154 27			NT				56.86 35.71 44.44						
	1960, [55]	Kafr El-Sheikh		*Argas refiexus hermanni*	1400		Nymph: 14 pools Adult: 14	VI HAI CF NT			7.14 7.14	7.14 7.14			7.14 7.14			tick
	1968, [26]	Alexandria		Human	120	60 60	3–13	CF HAI							4.00	4.00		Acute blood samples from children
	1969, [33]	Throughout Egypt		Human	1113	1113 (male)		HAI						51.21				University students Abstract
	1984, [24]	Giza (from Cairo or surrounding areas)		Human	55			VI HAI			0.00			1.82 (IgM) 58.18 (IgG)				Serum samples of non-specific fever and myalgia patient
	1989, [40]	Sharqiya		Human	437	215 222	8–14	ELISA	2.32 4.50									School children
	NA (1993 ^a^), [41]	Nile River Valley		Human	915			ELISA	20.00									Abstract
	1993, [56]	Aswan		*An. multicolor* *An. pharoensis* *An. tenebrosus* *Cx. antennatus* *Cx. perexiguus* *Cx. pipiens* *Cx. poicilipes* *Ochlerotatus caspius* *Urranotaenia unguiculata*	5 145 245 2691 9011 6982 26 16,889 30 (up to 88/pool)			VI			0.00 0.00 0.00 1.90 2.60 0.30 0.00 0.00 0.00 0.00 0.00 0.00							Minimum infection rate
	1999, [47]	NA		Human	264			ELISA	54.14									Workers of sewage treatment plants, Abstract
	1999–2002, [57]	Upper Egypt (Qena) Middle Egypt (Fayoum) Lower Egypt (Sharqiya) North Sinai South Sinai		Human	2203 1593 1292 202 675			ELISA	35.00 27.31 13.78 0.99 6.67									
		Upper Egypt (Qena) Middle Egypt (Fayoum) Lower Egypt (Sharqiya) North Sinai (Al-Areesh), South Sinai (Nuweiba)		*Culex* (81%) *Culex* (99%) *Aedes* (56%) *Culex* (37%) *Culex* (97%) *Culex* (99%)	880 (216) 9398 (484) 40,937 (1686) 34,770 (1897) 26,170 (731)			RT-PCR								0.00 0.00 0.18 0.58 0.14		*Cx. pepiens*, *Cx. perexiguus* *Cx. perexiguus*, *Cx. antennatus*, *Cx. poicilipes*, *An. pharoensis*, sand files *Cx. antennatus*
	2013–2014, [58]	Throughout Egypt		Human	160			ELISA RT-PCR	55.00							0.00		Blood donors from the blood bank of Ain Shams University Hospitals
	2018–2019, [59]	Qalyoubiya, Menoufiya, Kafr El-Sheikh, Gharbiya		Cattle Buffalo Camel Sheep Goats	100 50 50 85 75			ELISA	18.00 0.00 40.00 3.53 5.33									
	2019, [60]	Qalyoubiya Menoufia Kafr El-Sheikh Gharbiya		Horse	90 120 160 130			ELISA RT-PCR	25.56 17.50 21.25 14.62							0.00 0.00 0.63 0.00		
		Kafr El-Sheikh Gharbiya		*Cx*. spp.	5 pools				RT-PCR							20.00		
SINV	1968, [26]	Alexandria		Human	120	60 60	3–13	CF HAI						4.00	4.00			Acute blood samples from children
	1969, [33]	Throughout Egypt		Human	1113	1113 (male)		HAI						6.47				University students Abstract
	1993, [56]	Aswan		*An. multicolor* *An. pharoensis* *An. tenebrosus* *Cx. antennatus* *Cx. perexiguus* *Cx. pipiens* *Cx. poicilipes* *Oc.* (=*Aedes*) *caspius* *Ur. unguiculata*	5 145 245 2691 9011 6982 26 16,889 30 (up to 88/pool)			VI			0.00 0.00 0.00 0.00 0.30 0.00 0.00 0.00 0.00 0.00 0.00 0.00							
	1999, [47]	NA		Human	264			ELISA	1.13									Workers of sewage treatment plants Abstract
CHIKV	1984, [24]	Giza (from Cairo or surrounding areas)		Human	55			VI HAI			0.00			0.00 (IgM) 5.45 (IgG)				Serum samples of non-specific fever and myalgia patient

Abbreviations: AGPT: agar gel precipitation test; CHIKV: Chikungunya virus; CF: complement fixation test; DENV: dengue virus; HAI: haemagglutination inhibition test; IF: immunofluorescence assay; NA: not applicable to the field; NT: neutralization test; Ref.: reference; RVFV: Rift Valley fever virus; SINV: Sindbis virus; VI: virus isolation; WNV: West Nile virus; ^a^ shows the published year.

**Table 2 viruses-14-01577-t002:** Precision and risk of bias assessment for mosquito-borne virus prevalence measures in Egypt.

Virus	Author, Year	Region/City/Governorate	Species	Sampling Approach	Risk of Bias Assessment		Precision	Ref.
					Sampling	Assay		
DENV	Hussen, 2020	Upper Egypt (Sohag and Assiut)	Human Camel	MSCS MSCS	Low	High	Low	[34]
DENV, SINV, WNV	Mohammed, 1970	Alexandria	Human	Conv.	High	High	High	[26]
DENV, SINV, WNV	Darwish, 1975	Throughout Egypt	Human	CS	Low	Low	High	[33]
RVFV	Mroz, 2017	Nile Delta	Cattle	MSCS	Low	Low	High	[11]
	Mroz, 2017	Throughout Egypt	Different livestock	MSCS	Low	Low	High	[10]
	El-Bahgy, 2018	From different herds in Egypt and originated from Sudan	Camel	MSCS	Low	High	High	[53]
	Meegan, 1979	Qalyoubiya, Sharqiya, Giza	Human	Conv.	High	Low	Low	[36]
	Abdel-Rahim, 1999	Aswan and Assiut	Sheep Cattle	Conv. Conv.	High High	Low Low	Low Low	[45]
	Imam, 1979	Qalyoubiya, Sharqiya, Giza, Menia, Aswan, Sohag, Cairo	Different animals	Conv.	High	Low	High	[35]
	Hanafi, 2011	Aswan, Beni Suef, Behera, Cairo, Daqahliya, Kafr El-Sheikh, Qalyoubiya, Qena, and Sharqiya	Human	Conv.	High	Low	High	[50]
		Kafr El-Sheikh	Different livestock	Cluster	Low	High	High	
		Kafr El-Sheikh	Mosquito	Cluster	Low	Low	High	
	Youssef, 2001	Behera Alexandria Menia	Rat	Cluster	Low	High	High	[48]
	Youssef, 2002	Behera Alexandria Menia	Rat	Cluster	Low	Low	High	[49]
	Niklasson, 1979	Swedish soldiers serving in Egypt	Human	Cluster	Low	High	High	[37]
	Mahmoud, 2021	Aswan, Qena, and Luxor	Different livestock	Cluster	Low	High	Low	[9]
	Horton, 2014	Giza (originated from governorates throughout Egypt)	Different livestock	Cluster	Low	High	High	[52]
	Turkistany, 2001	Slaughterhouse personnel in Makkah from Egypt	Human	Cluster	Low	High	Low	[46]
	Abu-Elyazeed, 1996	Behera, Beni Suef, Cairo, Damietta, Daqahliya, Gharbiya, Giza, North Sinai, Ismailia, Kafr El-Sheikh, Menoufiya, Port Said, Qalyoubiya, Sharqiya, South Sinai	Human	Cluster	Low	High	High	[42]
	Youssef, 2009	Alexandria	Pig Human	Cluster Cluster	Low Low	High High	High Low	[51]
	Allam, 1986	Nile Delta	Sheep	Cluster	Low	Low	High	[38]
	Arthur, 1993	Aswan	Different livestock	Conv.	High	Low	Low	[44]
	Botros, 1988	Daqahliya	Sheep	Cluster	Low	Low	High	[39]
	Centers for Disease Control & Prevention (CDC), 1994	Aswan	Human	Cluster	Low	High	High	[43]
RVFV, SINV	El-Esnawy, 2001	NA	Human	Cluster	Low	High	High	[47]
RVFV, WNV, CHIKV	Darwish, 1987	Giza	Human	Conv.	High	Low	Low	[24]
RVFV, WNV	Corwin, 1992	Sharqiya	Human	MSCS	Low	High	High	[40]
RVFV, WNV	Corwin, 1993	Nile River Valley	Human	Random	Low	High	High	[41]
SINV, WNV	Turell, 2002	Aswan	Mosquitoes	Random	Low	Low	High	[56]
WNV	Selim, 2020	Qalyoubiya, Menoufia, Kafr El-Sheikh, Gharbia	Horse	Random	Low	Low	High	[60]
	Selim, 2020	Qalyoubiya, Menoufia, Kafr El-Sheikh, Gharbia	Different livestock	Random	Low	High	High	[59]
	Soliman, 2010	Upper Egypt (Qena), Middle Egypt (Fayoum), Lower Egypt (Sharqiya), North Sinai (Al-Areesh), South Sinai (Nuweiba)	Human	Random	Low	High	High	[57]
	Youssef, 2017	Blood donors from the blood bank of Ain Shams University Hospitals	Human	Random	Low	Low	High	[58]
	Schmidt, 1964	Kafr El-Sheikh	Tick	MSCS	High	Low	High	[55]
	Schmidt, 1963	Upper and Lower Egypt	Equine	Random	Low	Low	High	[54]

Risk of bias (ROB) assessment was considered low when (i) sampling was based on random or cluster selection and (ii) measures included viral neutralization testing or biological assays (i.e., virus isolation or RT-PCR). ROB was considered high for (i) acute infection studies or studies on sera or tissues taken from the affected herds, or (ii) measures conducted by immunological assays. Studies were considered to have a high precision if the number of individuals tested was ≥100 for humans and animals and ≥1000 for vectors. Abbreviations: CHIKV: Chikungunya virus; Conv: convenience sampling; CS: cluster sampling; DENV: dengue virus; MSCS: multi-stage cluster sampling; Ref: reference; RVFV: Rift Valley fever virus; SINV: Sindbis virus; WNV: West Nile virus.

## Data Availability

Not applicable.

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
