# Peer review of "Epidemiology of Mosquito-Borne Viruses in Egypt: A Systematic Review"

_viruses, 2022, doi:10.3390/v14071577_

Round 1
Reviewer 1 Report
Table 1 needs to be organised and formatted so it's easier to read. Same could be done with Table 2.
Well written and comprehensive analysis of the prevalence of mosquito borne diseases in Egypt.
Author Response
Table 1 needs to be o lyrganised and formatted so it's easier to read. Same could be done with Table 2.
R: For table 1, we deleted the number of positive samples after the positive rate, and combined the reference with the column of ‘year’. The length has been shortened to a third than before. For table 2, we deleted the columns of ‘Year(s)’ of Study, and the sample size, since these data can be found in table 1.
Reviewer 2 Report
This review describes “Epidemiology of mosquito-borne viruses in Egypt: a systematic review”.
The review intend to address important issue which might be utilized to improve medical care and public health not only in Egypt but also rest of the world with similar infections. The authors try to emphasize on the cause of MBVs and put forward their idea of seroprevelence studies which can improve in reduction of MBVs to an extent. The title selected seems too broad but the data presented in it is not up to the mark.
However, I do suggest certain things which need clarification to support and strengthen the Review article.
Abstract
Line 14: Please use common instead of historic.
Line 18: 3,23,1, 4 what are these numbers? If they represent number of publications considered for the referred viruses and writing this review, they are very less.
Line 19: Please mention the abbreviation West Nile virus for WVV.
Line 20-22: reframe the sentence to “The monophyletic characterisation of RVFV and WNV strains found in Egypt, which spans about 20 half a century, suggests that both RVFV and WNV are widely transmitted in this nation”.
Overall the abstract needs to be restructured to come up with informative and importance of the review.
Introduction
Line 31-35: Please shorten sentences (1-2 lines) related to COVID 19 which has no relevance to the topic selected.
Line 43: any reference or report on this topic.
Line 63: Ae. Aegypti. Please correct it to A. aegypti. Check throughout the text.
Methods:
Line 94: Do you use WHO reports in you studies in case not include them and provide references.
It could be better you first provide the general concept globally and then talk about Egypt cases and make possible correlation.
Line 99: “participants’ “ please place the comma and hypens in place.
This paragraph needs improvements regarding brief explanation of data extraction steps.
Better will be to elaborate and tabulate. It will provide readability and clearity.
Line 124: what is the meaning of 19,2 and 3? explain it briefly ( what animals, vectors).
Table 1:
The records presented in the table are very old and of less relevance if writing on updates on such topic. Please only consider reviews from last 10-15 years which can improve the table presentation. You can mention other information in text if it is of great relevance.
In present form the table is extensive and less informative. Please highlight some places which are more relevant to the subject.
Fig.1: how do you get three more records after screening?
These numbers are confusing (n=1, 1) what does it means? Explain in the legend.
Table 2: low/ high makes the things confuse please write in legends what does it mean?
Line 160: From this sentence what i understand is that around 60% of the work was done earlier to year 2000 now you have included 40% data. If that is the case why do you chose previous data you could have gone for data around 10-15 years for this publication. This will be of much significance, relevance to the topic.
Line 250: please do not repeat the sentences
Discussion
The discussion should be point specific. There are many places where it lack important information on the present subject. So I recommend please focus on discussion with most relevant references.
Conclusion
Conclusion: it could have been better you can present future prospective with recent pharmacological technologies applied.
General comments:
The reviews needs to be corrected by the native speaker or sent for English correction.
There are some places which loose its connection between sentences. Please verify them.
Improvement in the quality of review is must to reach the standards of the publication.
Please once again correct the typos and keep care before submission.
Please provide the graphical abstract to present this topic effectively.
The manuscript needs some serious corrections, I still ask the authors to rearrange and extend information on certain parts of the manuscript especially in the introduction and discussion with additional references.
I do think that the manuscript contains important issues, information, interesting approaches, which can lead to proper understanding of MBVs for improving medical care and public health in Egypt and worldwide.
Author Response
Abstract
Line 14: Please use common instead of historic.
R: Done as suggestion.
Line 18: 3,23,1, 4 what are these numbers? If they represent number of publications considered for the referred viruses and writing this review, they are very less.
R: We have deleted these numbers in the current version. The number of researches on the prevalence of mosquito-borne viruses Egypt, which met the criteria of the search strategy was less, reflecting the deficiency of MBV studies in Egypt. More than half of them were performed before 2000, especially for SINV and CHIKV. The retrospect analyses improve our understanding on the transmission dynamics of MBVs in Egypt, especially on the RVFV, DENV, and WNV. It can arise public attention on MBVs, and essential to support the decision-making process regarding appropriate prevention and control strategies on MBVs in Egypt.
Line 19: Please mention the abbreviation West Nile virus for WVV.
R: We have checked and provided the full names of all the abbreviations in the revised version.
Line 20-22: reframe the sentence to “The monophyletic characterisation of RVFV and WNV strains found in Egypt, which spans about 20 half a century, suggests that both RVFV and WNV are widely transmitted in this nation”.
R: Done as suggestion.
Overall the abstract needs to be restructured to come up with informative and importance of the review.
Introduction
Line 31-35: Please shorten sentences (1-2 lines) related to COVID 19 which has no relevance to the topic selected.
R: Done as suggestion.
Line 43: any reference or report on this topic.
R: We have added relative reference in the revised version.
Line 63: Ae. Aegypti. Please correct it to A. aegypti. Check throughout the text.
R: Ae. aegypti is more commonly used in literatures than A. aegypti. We have checked that all the species names are in italics throughout the text.
Methods:
Line 94: Do you use WHO reports in you studies in case not include them and provide references. It could be better you first provide the general concept globally and then talk about Egypt cases and make possible correlation.
R: We have checked WHO reports relates to mosquito-borne diseases in Egypt. However, most of them are annual case reports for the year of MBV outbreak (https://www.who.int/news-room/fact-sheets/detail/rift-valley-fever; http://apps.who.int/iris/bitstream/handle/10665/253767/EMROPUB_2016_EN_19264.pdf?sequence=1&isAllowed=y; http://www.emro.who.int/pandemic-epidemic-diseases/news/infectious-disease-outbreaks-reported-in-the-eastern-mediterranean-region-in-2018.html). No sample size, study region, and measures were in details.
Line 99: “participants’ “ please place the comma and hypens in place.
R: We have modified it as ‘age and sex of participants’.
This paragraph needs improvements regarding brief explanation of data extraction steps. Better will be to elaborate and tabulate. It will provide readability and clearity.
R: More details were added in the text to make the data extraction steps more clearly.
Line 124: what is the meaning of 19,2 and 3? explain it briefly ( what animals, vectors).
R: We added specific species after animals, and vectors.
Table 1:
The records presented in the table are very old and of less relevance if writing on updates on such topic. Please only consider reviews from last 10-15 years which can improve the table presentation. You can mention other information in text if it is of great relevance.
In present form the table is extensive and less informative. Please highlight some places which are more relevant to the subject.
R: As we mentioned before, the number of researches on the prevalence of mosquito-borne viruses Egypt, which met the criteria of the search strategy was less, reflecting the deficiency of MBV studies in Egypt. And more than half of them (19/33) were conducted before 2000. Moreover, all the literatures on SINV and CHIKV were performed on last century. Although these records are old, they can help us to understand the transmission dynamics of MBVs in Egypt. For table 1, we deleted the number of positive samples after the positive rate, and combined the reference with the column of ‘year’.
Fig.1: how do you get three more records after screening?
R: Thanks for your kind suggestion. It was one slip of the pen. It has been corrected in the revised version.
These numbers are confusing (n=1, 1) what does it means? Explain in the legend.
R: Done as suggestion.
Table 2: low/ high makes the things confuse please write in legends what does it mean?
R: We added the criteria of Risk of bias assessment in brief in the legend of Table2.
Line 160: From this sentence what i understand is that around 60% of the work was done earlier to year 2000 now you have included 40% data. If that is the case why do you chose previous data you could have gone for data around 10-15 years for this publication. This will be of much significance, relevance to the topic.
R: As we mentioned above, the combination on studies of before and after 2000 can help us to understand the transmission dynamic of MBVs in Egypt. And for the lack of prevalence studies on MBVs in Egypt, if we removed the studies before 2000, the number of researchers involved in following epidemiological study will be too less, and the data on SINV and CHIKV will be in blank.
Line 250: please do not repeat the sentences
R: We mentioned the two imported cases in Russia from Egypt in the introduction. Here, we would like to emphasize the serious status of DENV outbreak in the Red Sea region, even the spillover event to other countries have occurred. The two exported cases from Egypt to Russia were happened in 2017, not the same spillover event of DENV from Egypt occurred in 2009/2010 in Italy.
Discussion
The discussion should be point specific. There are many places where it lack important information on the present subject. So I recommend please focus on discussion with most relevant references.
R: Since we hope that the review could give the readers a more comprehensive understanding of the MBV prevalence in Egypt, the way of their transmission, possible imported routes, the potential reasons on resurgence of DENV, and RVFV, current prevention strategy (the efficiency of vaccination), and the pressure of outbreaks of MBVs in neighboring countries on MBV control in Egypt.
Conclusion
Conclusion: it could have been better you can present future prospective with recent pharmacological technologies applied.
R: Sorry that I am not so clear the ‘pharmacological technologies’, you mentioned here, since that there’s no knowledge on therapy methods of MBVs in the text. If you mean the prospective of the prevention strategy we proposed in the manuscript, it has been mentioned in the text that the systematic surveillance on MBVs, can help to estimate the accurate disease burden, evaluate the risk of endemic transmission of MBVs, and establish an early warning system of MBV outbreak in Egypt.
General comments:
The reviews needs to be corrected by the native speaker or sent for English correction.
R: The language of the manuscript has been improved by ‘Editage’.
There are some places which loose its connection between sentences. Please verify them.
R: Done as suggestion.
Improvement in the quality of review is must to reach the standards of the publication.
R: Done as suggestion.
Please once again correct the typos and keep care before submission.
R: Done as suggestion.
Please provide the graphical abstract to present this topic effectively.
R: The graphical abstract we provided includes two figures. The left one shows the subject category, geographic coverage, and time rage of epidemiological of MBV performed in Egypt. The right one shows that monophyletic characterization of RVFV strains prevalence in Egypt. These are two main issues we would like to arise public attention in this review, the scarcity of MBV surveillance based on pathogen detection, and the frequency of MBV circulation in the nation.
Reviewer 3 Report
Epidemiology of mosquito-borne viruses in Egypt: a systematic review
This is an interesting and comprehensive compilation on mosquito-borne diseases in Egypt and their current situation. This manuscript shows the gaps in currently available information on mosquito-borne diseases in Egypt and the steps to solve that problem.
I believe that having all the required information facilitates decision-making for vector control authorities, which is the most substantial contribution of this manuscript. I think this manuscript deserves to be published in Viruses.
I have only some minor suggestions:
♠ Please have an English language expert polish your writing.
♠ Line 18. It would be better to have those studies in ascending or descending order
♠ Line 19. The authors describe only some abbreviations of the virus names. Please describe all of them or none.
♠ Please remember to write each scientific name in italics throughout the text (lines 57, 64, 290, 325, 326, among many others).
♠ Lines 401-402. It is confusing to have too many numbers together. Please move the reference numbers to the end of the sentence.
Author Response
♠ Please have an English language expert polish your writing.
R: The language of the manuscript has been improved by ‘Editage’.
♠ Line 18. It would be better to have those studies in ascending or descending order
R: In the revised version, we have reorganized the abstract, and this sentence has been deleted.
♠ Line 19. The authors describe only some abbreviations of the virus names. Please describe all of them or none.
R: Done as suggestion.
♠ Please remember to write each scientific name in italics throughout the text (lines 57, 64, 290, 325, 326, among many others).
R: We have checked and modified this issue throughout the text.
♠ Lines 401-402. It is confusing to have too many numbers together. Please move the reference numbers to the end of the sentence.
R: Done as suggestion.
Round 2
Reviewer 2 Report
Nearly all of the issues I highlighted with this Manuscript have been addressed. I accept the review article for publication in present form.